# Technical Note: Algal Pigment Index 2 in the Atlantic off the Southwest Iberian Peninsula: standard and regional algorithms

Priscila Costa Goela[1,2], Sónia Cristina[1,2], Tamito Kakiyama[3], John Icely[1,2], Gerald Moore[4], Bruno Fragoso[1,2], Alice Newton[1,5]

[1]Centre for Marine and Environmental Research, FCT, University of Algarve, Campus de Gambelas, 8005-139 Faro, Portugal
[2]Sagremarisco Lda., Apartado 21, 8650-999 Vila do Bispo, Portugal
[3]University of Algarve, Campus de Gambelas, 8005-139 Faro, Portugal
[4]Bio-Optika, Crofters, Gunnislake, UK
[5]Norwegian Institute for Air Research-IMPEC, Box 100, 2027 Kjeller, Norway

*Correspondence to*: Priscila Costa Goela (priscila.goela@gmail.com)

**Abstract.** In this study, Algal Pigment Index 2 (API2) is investigated in Sagres, an area located in the Atlantic off the southwestern Iberian Peninsula. Standard results provided by MEdium Resolution Image Spectrometer (MERIS) ocean colour sensor were compared with alternative data products, determined through a regional inversion scheme, using both MERIS and in situ remote sensing reflectances ($R_{rs}$) as input data. The reference quantity for performance assessment is in situ total chlorophyll *a* (TChl*a*) concentration estimated through phytoplankton absorption coefficient (i.e., equivalent to API2). Additional comparison of data products has also been addressed for TChl*a* concentration determined by High Performance Liquid Chromatography. The MERIS matchup analysis revealed a systematic underestimation of TChl*a*, which was confirmed with an independent comparison of product maps analysis. The study demonstrates the importance of regional algorithms for the study area that could complement upcoming standard results of the current Sentinel-3/OLCI space mission.

Keywords: absorption, MERIS Algal Pigment Index 2, Multi-Layer Perceptron (MLP) neural nets, ocean colour, remote sensing.

## 1 Introduction

The MERIS space sensor, operated by the European Space Agency (ESA) on-board of the Envisat platform from 2002 to 2012, has been continuously supported by investigations for the assessment and improvement of data products. Commissioned studies include the validation of radiometric data such as the $R_{rs}$ (Cristina et al., 2014; Kajiyama et al., 2014), as well the analyses of derived product maps (Kajiyama et al., 2014; D'Alimonte et al., 2014; Cristina et al., 2016). These MERIS validation activities have established an important basis to address Earth Observation (EO) capabilities through the Ocean Land Colour Instrument (OLCI) sensor launched on the Sentinel-3 satellite in February 2016. OLCI data products are

the main component of the Copernicus European programme to monitor the marine environment, and the retrieval of Chlorophyll a (Chl*a*) is a core task of the Sentinel-3 space mission. Chl*a* is needed to estimate the phytoplankton biomass in the ocean and to contribute to a variety of inter-related investigations and applications, including climate data records, environmental legislation, and a number of economic activities such as fisheries and aquaculture. After the removal of the

atmospheric contribution to the signal recorded at the top of the atmosphere, Chl*a* can be estimated from the bottom-of-atmosphere (BOA) $R_{rs}$ values using the standard approach with polynomial algorithms based on band-ratios of the input radiometric quantities. The corresponding MERIS data product is denoted Algal Pigment Index 1 (API1) (Morel and Antoine, 2011). The use of band-ratio is based on the assumption that seawater optical properties are driven by Chl*a*. A tendency to overestimation has however been documented in optically complex marine conditions (D'Alimonte et al., 2014).

This can occur when optically active constituents such as Coloured Dissolved Organic Matter (CDOM) and detrital particulate matter exceed their typical levels. The Chl*a* retrieval accuracy declines in these optically complex conditions because the band-ratio approach attributes variations of the $R_{rs}$ spectral slope to changes of Chl*a*. In such cases, regionalized bio-optical algorithms are required (Bricaud et al., 2002, Gregg and Casey, 2004). Alternative ocean colour inversion schemes adopted to improve the Chl*a* retrieval from space include artificial Neural Nets (NN) using $R_{rs}$ at selected

wavelengths as input. In the case of MERIS standard deliverables, this corresponds to the API2 data product (Doerffer and Schiller, 2007).

Although NNs can in principle model any relationship between apparent and inherent optical properties, their performance is in practice mostly determined by the dataset used for their training. Specific analyses are then needed to compare the standard MERIS API2 results with independent estimates. This main requirement is addressed in the present work by: 1)

developing and assessing the performance of an independent regional Multilayer Perceptron (MLP) scheme to retrieve results equivalent to MERIS API2 values; and 2) comparing MERIS standard and regional API2 product maps.

The region under study is the Atlantic off the Southwestern Iberian Peninsula, where in situ reference data were collected at three stations off the Sagres region at 2, 10 and 18 km from the coast (henceforth stations A, B and C, respectively). The study is conducted based on both matchup analyses and product map inter-comparisons, with timely presentation of the

results acknowledging, not only the planned MERIS data reprocessing, but also the need for a benchmark for the analysis of the upcoming OLCI API2 deliverables. An added value of this study is to confirm that qualitative evaluations based on product maps comparison can complement matchup data at the early mission stages of OLCI, when the statistical significance of matchup analysis is limited.

## 2 Data and Methods

Field campaigns were performed from 2008 to 2012 at the three study sites, with simultaneous collection of water samples and radiometric measurements. MERIS Level 2 Full Resolution (FR, 290 m x 260 m) and Reduced Resolution (RR, 1.20 km x 1.04 km) satellite images were extracted for matchup analysis and product map comparison, respectively, and analyzed

with the Basic ERS & ENVISAT (A) ATSR and MERIS Toolbox (BEAM version 4.9). The MEGS 8.1 processor (MERIS third reprocessing) was used to derive level 2 data, in agreement with previously reported extraction procedures (Cristina et al., 2014, 2015). The selection of satellite images was restricted to images without clouds and contamination, as indicated by not having specific Product Confidence (PCD), sun glint and ice flags. More details on the image selection criteria and full

description of flags are reported in Cristina et al. 2016. TChl$a$ concentration (monovinyl Chl$a$ + divinyl Chl$a$ + chlorophyllide $a$ + phaeopigments) was determined by High Performance Liquid Chromatography (HPLC), according to Wright and Jeffrey (1997), herein referred to as TChl$a_{\mathrm{HPLC}}^{\mathrm{REF}}$. The protocols adopted for TChl$a$ extraction, identification and quantification procedures are reported in Goela et al. (2014, 2015).

## 2.1 In situ reference data

In situ radiometric measurements were acquired with a tethered attenuation coefficient chain sensor (TACCS, Satlantic®), supporting a hyperspectral surface irradiance sensor $E_{\mathrm{s}}(\lambda)$ and a subsurface radiance sensor $L_{\mathrm{u}}(\lambda)$, as well as a tethered attenuation chain equipped with four irradiance sensors at nominal depths of 2, 4, 8 and 16 m. Normalized water leaving reflectance ($\rho_N$) was computed with Eq. (1):

$$\rho_N(\lambda) = \pi \frac{L_W(\lambda)}{E_S(\lambda)}, \tag{1}$$

where $L_{\mathrm{w}}$ is the water leaving radiance determined by propagating $L_{\mathrm{u}}$ from below to above the sea surface and corrected for self-shading following (Gordon and King, 1992). $\rho_N(\lambda)$ corresponds to the remote sensing reflectance $R_{\mathrm{rs}}$ upon scaling with π.

For the determination of in situ absorption of phytoplankton pigments at 442 nm ($a_{\mathrm{ph}}(442)$), seawater filtrates (0.5 L) were collected on GF/F filters (pore size 0.7 μm), which were then analyzed with the transmittance-reflectance technique as in

(Tassan and Ferrari, 2002), using a dual beam-spectrophotometer (GBC® CINTRA 40), equipped with an integrating sphere. The phytoplankton absorption was determined as the difference between the total particulate and detrital absorption, which were measured before and after sodium hypochlorite bleaching (Ferrari and Tassan, 1999; Goela et al., 2013), respectively. The API2 in situ equivalent algal pigment index TChl$a_{\mathrm{ABS}}^{\mathrm{REF}}$ was then estimated by converting $a_{\mathrm{ph}}(442)$ into API2, using the same regression coefficients presented in Sect. 2.2.2.

## 2.2 Chlorophyll $a$ retrieval algorithms

### 2.2.1 MERIS Standard algorithm API2

This standard product is estimated with two NNs. The first NN computes BOA $R_{\mathrm{rs}}$ values by removing the atmospheric radiometric contribution from input space-born $R_{\mathrm{rs}}$ values. The second NN utilizes the BOA $R_{\mathrm{rs}}$ to derive the $a_{\mathrm{ph}}(442)$. The final API2 product is then computed as $\mathrm{MER}^{\mathrm{API2}} = A \times a_{\mathrm{ph}}(442)^{\mathrm{B}}$, with power-law regression coefficients A=21.0 and

B=1.04 derived from field measurements in the German Bight and Norwegian waters (Doerffer and Schiller, 2007).

### 2.2.2 Regional MLP NN algorithm

The regional MLP for retrieving the data product equivalent to API2 has been trained with the in situ data collected at the Sagres site (instructions for independent implementation by users are provided as supplemental material http://ocportugal.org/sites/default/files/mlpSgrAPI2.pdf). This MLP is here applied to two different sets of input data for assessment of performance and for comparison of results. The first set consists of the in situ $R_{rs}$ values ($R_{rs}^{SITU}$), and the second set includes standard MERIS BOA $R_{rs}$ data ($R_{rs}^{MER}$). Corresponding data products are denoted MLP($R_{rs}^{SITU}$) and MLP($R_{rs}^{MER}$), respectively. In both cases, $R_{rs}$ at 490, 510 and 560 nm were selected as input channels, in agreement with the reference study (Cristina et al., 2014).

A novelty detection scheme (D'Alimonte et al., 2014; Bishop, 1994) was used to verify the algorithm applicability range by evaluating the representativeness of the input data in the training dataset (D'Alimonte et al., 2003; Mélin et al., 2011; Sá et al., 2015). The adopted applicability range is based on a novelty index ($\eta$) presented in published works (D'Alimonte et al., 2013; Sá et al., 2015). A revision is however applied for the scope of this work. This updated version considers all dimensions of the Principal Component Analysis (PCA) of selected input data, rather than only the first three components considered in the past (see the supplemental material for details). This updated definition is more effective for cases where the variability of training and application data tends to occur at different wavelengths (details not presented here). Key features are: 1) $\eta$ is bounded between 0 and $\infty$; 2) the more the $R_{rs}$ spectrum is similar to the in situ MLP training measurements, the lower is its $\eta$; and 3) an $R_{rs}$ spectrum is considered within the MLP applicability range when $\eta \leq 1$.

### 3 Results

The main tasks of this study are: 1) to evaluate the performance of regional MLP algorithm and the MER$^{API2}$ results with respect to the in situ TChl$a_{ABS}^{REF}$ reference measurements; 2) to verify the applicability of the regional MLP($R_{rs}^{MER}$) and to compare product maps with MER algal pigment indices; and 3) to extend the analysis by also considering TChl$a_{HPLC}^{REF}$ for data product assessment.

The statistical figures used to evaluate the estimated ($y$) in relation to the reference in situ TChl$a$ ($x$), were absolute ($\varepsilon$) and signed ($\delta$) percent differences, defined as:

$$\varepsilon = \frac{1}{N}\sum_{i=1}^{N}\frac{|y_i-x_i|}{x_i} \times 100; \quad \delta = \frac{1}{N}\sum_{i=1}^{N}\frac{y_i-x_i}{x_i} \times 100 , \qquad (2)$$

where $N$ is the total number of samples and $i$ is the sample index. For product maps comparison, the absolute ($\varepsilon^*$) and signed ($\delta^*$) unbiased differences were instead determined as:

$$\varepsilon^* = \frac{1}{N}\sum_{i=1}^{N}\frac{|y_i-x_i|}{y_i+x_i} \times 200; \quad \delta^* = \frac{1}{N}\sum_{i=1}^{N}\frac{y_i-x_i}{y_i+x_i} \times 200 , \qquad (3)$$

where $x_i$ and $y_i$ are the MLP($R_{rs}^{MER}$) and MER$^{API2}$ values, respectively, taking the mean of the two values as a reference. In addition, the coefficient of determination $r^2$ between the evaluated quantities is also reported. The total number of samples used to validate MER$^{API2}$ and MLP($R_{rs}^{MER}$) algorithms results with respect to the in situ reference measurements was N=54. In contrast, the total number of samples for assessing the performance of regional MLP algorithm with in situ reference measurements (MLP($R_{rs}^{SITU}$), was N=297. This larger number of samples is based on the data from 4-8 radiometric casts for each in situ TChla sample at each location.

## 3.1 Matchup data analysis

The top panels of Fig. 1 present the matchup comparisons of MER$^{API2}$, MLP($R_{rs}^{MER}$) and MLP($R_{rs}^{SITU}$) with respect to the in situ reference TChl$a_{ABS}^{REF}$ (Figs. 1a, 1b and 1c, respectively). While MER$^{API2}$ underestimated TChl$a$ ($\delta$ = -34%) especially at higher concentrations, the regional products slightly overestimated TChla ($\delta$ =11% for MLP($R_{rs}^{MER}$) and 2% for MLP($R_{rs}^{SITU}$). The best agreement between data sets was obtained with MLP($R_{rs}^{SITU}$), while MER$^{API2}$ showed larger uncertainties. Table 1 presents the matchup analysis where the underestimation of MER$^{API2}$ in relation to TChl$a$ is relatively constant (35%, 32% and 34%, in stations A, B and C, respectively) in all stations, but the correlation coefficient improves with distance offshore (0.22, 0.60, 0.67 in stations A, B and C, respectively).

In general, the matchup analysis with TChl$a_{HPLC}^{REF}$ revealed higher uncertainties for MER$^{API2}$, MLP($R_{rs}^{MER}$) and MLP($R_{rs}^{SITU}$), as detailed in Fig. 1 (lower panel). Note that also in this case MLP($R_{rs}^{SITU}$) presented the best results, with the highest coefficient of determination and the lowest bias. Similar to what was documented for TChl$a_{ABS}^{REF}$, the bias for TChl$a_{HPLC}^{REF}$ displayed only small differences between the sampling stations. The coefficient of determination instead increased from station A to station C. The underestimation of MER$^{API2}$ in relation to TChl$a_{HPLC}^{REF}$ was also observed, but with a lower bias (Fig. 1d). These observations are schematized in Fig. 2, where MER$^{API2}$ was considered as the baseline. A complementary comparison with MER$^{API1}$ is also presented for completeness. Results indicated an overestimation by the API1 algorithm in relation to both estimations of TChl$a$ (details not shown). The tendency of TChl$a_{ABS}^{REF}$ to produce higher values than TChl$a_{HPLC}^{REF}$ was also confirmed.

## 3.2 Comparison of product maps

The comparison of MERIS API2 standard product with the MLP regional results is presented on Fig. 3. The maps for the regional MLP (Fig. 3a) and the MER$^{API2}$ (Fig. 3b) are shown in the top panel, together with the difference between MER$^{API2}$ and MLP($R_{rs}^{MER}$) shown in Fig. 3c. Overestimations of more than 35% in relation to the regional MLP are coloured in pink, and underestimations below 35 % are coloured in yellow, while differences between -35% and 35% are in green. The MLP($R_{rs}^{MER}$) region of applicability is shown in Fig. 3d, with black contours indicating the threshold η=1. Results indicate an underestimation by MER$^{API2}$ of more than 35% in a significant part of the applicability range, especially near the coast.

The results from the application of Sagres regional MLP to the Atlantic off the Portuguese coast is presented in Fig. 3e and Fig. 3f. Besides the Sagres area (#3, in blue), two other regions of interest (ROIs) were chosen for comparison of product maps: Figueira da Foz (#1, in red) and Lisbon region (#2, in green, Fig. 3e). Note that ROI #1 and #2 have been selected for their contrasting features: the first influenced by Mondego river plume and the second by the Tagus estuary. The comparison between the MER$^{API2}$ and regional MLP products is presented as a scatter plot (Fig. 3f), following the same colour coding of the three ROIs. The underestimation tendency of MER$^{API2}$ in relation to in situ TChl$a$ was confirmed through this analysis. The results also indicated more pronounced differences in Mondego and Tagus ROIs, where values of TChl$a$ were higher. The statistical figures of the product map comparison between MER$^{API2}$ and regional MLP are summarized in Table 2. The applicability of the Sagres MLP was verified with the novelty detection scheme. The number of total and valid (i.e., $\eta < 1$) data points are denoted as $N_{tot}$ and $N_{val}$, respectively. The Sagres ROI presents the highest number of valid data points, while Tagus region had the highest percentage of novel data points.

## 4 Discussion

This study analyzed the standard MERIS API2 product by considering the TChl$a$ retrieval in the coastal waters of Portugal. Data product comparisons have been performed by developing and applying a regional MLP trained with Sagres in situ data and accounting for its applicability range. The work highlighted a tendency of MER$^{API2}$ to underestimate TChl$a$, not only when the reference values were derived through $a_{ph}(442)$ but also when determined by HPLC. This result is consistent with other studies addressing low productivity waters (Tilstone et al., 2012). This underestimation tendency is more pronounced at higher concentrations but not observed in the results of the regional MLP. Possible explanations can be uncertainties in BOA $R_{rs}$ values, as well as in specific properties of the NN inversion scheme used to compute the standard API2 values. It is noted that the MERIS NN scheme for API2 retrieval is scoped for global applications in both Case 1 and optically complex waters. This general applicability might limit the algorithm performance in the presence of specific bio-optical relationships at the regional scale. An example could be the upwelling along the coast of Portugal (Loureiro et al., 2005; Goela et al., 2015).

As a contribution to the forthcoming OLCI mission, the present work also provides indications to enhance standard OLCI API2 results by including additional training samples in the synthetic dataset used for the development of the MERIS NN scheme. The overestimation of TChl$a_{ABS}^{REF}$ in relation to TChl$a_{HPLC}^{REF}$ has been identified in this study as one of the reasons for the systematic difference observed in the comparison of MER$^{API2}$ with both in situ referred targets (Fig. 2b).

The regional MLP using in situ $R_{rs}$ as input produced highly accurate results (bias of 2%), when relating $R_{rs}^{SITU}$ to reference measurements of TChl$a_{ABS}^{REF}$. When MERIS $R_{rs}$ is used, the bias is slightly higher, probably due to the uncertainties of the atmospheric correction (Cristina et al., 2014). It is also reported that a cross-validation analysis performed by splitting the in situ data in different subsets to develop and assess the regional MLP documented an increase from 2 to 9% of the bias (details not presented). As observed for the standard NN inversion schemes, the performance of the regional MLP could be

enhanced through a better representation of the optical properties of the study region: the collection of additional field measurements is hence recommended. Another aspect that has been considered is the reduction in bias when the training dataset was $TChla_{ABS}^{REF}$ estimated with $a_{ph}$ at 440 nm (7% of bias). This indicates that the specific selection of the wavelength of the maximum phytoplankton absorption could allow for a better TChl$a$ parameterization and hence also lead to a more accurate regional MLP.

The strong relationship between $R_{rs}$ and the phytoplankton coefficient of absorbance at 442 nm suggests the presence of Case 1 waters. The better agreement with $TChla_{ABS}^{REF}$ rather than with $TChla_{HPLC}^{REF}$ can be explained by considering that the training of the neural net was performed with $TChla_{ABS}^{REF}$. An additional explanation could be that $TChla_{ABS}^{REF}$ was determined using $a_{ph}(442)$, which is likely better related to $R_{rs}$ than $TChla_{HPLC}^{REF}$ (both $a_{ph}(442)$ and $R_{rs}$ directly represent optical properties). A caveat would however apply to this argument, whereas $TChla_{HPLC}^{REF}$ is a direct measurement of the TChl$a$ concentration, $TChla_{ABS}^{REF}$ is an indirect measurement which has errors associated with the laboratory determination of $a_{ph}(442)$.

It is also noted that the regional relationship between $a_{ph}$ at 442 nm and TChl$a$ retrieved by HPLC is close to that used in MER$^{API2}$ (TChl$a_{MERIS} = 21\ a_{ph}(442)^{1.04}$, TChl$a_{SAGRES} = 27\ a_{ph}(442)^{1.13}$). However, the local relationship between TChl$a$ and $a_{ph}(442)$ corresponds to a coefficient of determination $r^2 = 0.8$. Hence, about 20% of variability of TChl$a$ is not related to $a_{ph}(442)$.

The ROIs data analysis indicates lower MERIS API2 values with respect to equivalent results derived with the regional MLP, especially when the TChl$a$ concentration increases. This finding is in a good agreement with the matchup results, thereby, highlighting the benefit of independent comparison of product maps to qualitatively evaluate data products at an early stage of ocean colour space missions (e.g., OLCI).

20 **5 Conclusions**

The scope of this technical note was to analyze the MERIS standard API2 product in the Southwestern coast of Portugal. A regional MLP algorithm to retrieve TChl$a$, estimated through phytoplankton absorption coefficient, was implemented and applied for this purpose. This regional algorithm produced good agreement with in situ data, hence indicating a high accuracy of regional MLP products. The applicability of the regional MLP in the study area was verified by a novelty

25 detection scheme. With this information, the study reports an underestimation tendency of MER$^{API2}$, which is consistent with other European basins within low ranges of this constituent. The results of the regional MLP were closer to the in situ reference for API2 – TChl$a$ estimated with $a_{ph}(442)$ – than to TChl$a$ determined by HPLC. This work also indicates that the use of a regional relationship between phytoplankton absorption and pigment concentration is expected to improve the accuracy of global ocean colour remote sensing products.

30 This study has highlighted the usefulness of maintaining in situ measurement programmes for validation purposes of ongoing ocean colour missions. Moreover, it has also demonstrated the importance of developing regional algorithms that

not only complement standard approaches, but that can also be applied for the qualitative data assessments of new ocean colour missions in the early stages of product map delivery (e.g., Sentinel-3).

**Data availability**

The majority of the in situ data used in this work can be accessed through the ESA MERIS MAtchup In-situ Database (http://mermaid.acri.fr/home/home.php) and the MERIS satellite data can be accessed through the Optical Data processor of ESA (http://www.odesa-info.eu/process_basic/basic.php).

**Acknowledgements**

The authors thank Dr. Davide D'Alimonte for his contribution to the MLP NN algorithm development and training, and wise advice both on the methodology design and in the interpretation of the results. This work was supported in part by the European Space Agency (ESA) for the "Technical Assistance for the Validation of MERIS Marine Products at Portuguese oceanic and coastal sites"(contract no. 21464/08/I-O) and "MERIS validation and algorithm 4th reprocessing" (contract no. ARG/003-025/14067Sagremarisco and ARG/003-025-1406/CIMA). Priscila Costa Goela and Sónia Cristina were funded by PhD grants from the Portuguese FCT(SFRH/BD/78356/2011 and SFRH/BD/78354/2011, respectively); Alice Newton was funded by EU FP7 project DEVOTES (grant no. 308392); John Icely is funded by EU FP7 AQUA-USER (grant no. 607325), and Horizon 2020 AquaSpace (grant no. 633476).

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

**Table 1. Comparison between standard (MER$^{API2}$), regional bio-optical algorithms (MLP($R_{rs}$$^{MER}$) and MLP($R_{rs}$$^{SITU}$)) and TChl$a$$^{REF}$**

| | N | | | | $\varepsilon$(%) | | | | $\delta$(%) | | | | $r^2$ | | | |
|---|---|---|---|---|---|---|---|---|---|---|---|---|---|---|---|---|
| | A | B | C | **All** | A | B | C | **All** | A | B | C | **All** | A | B | C | **All** |
| MER$^{API2}$ *vs* TChl$a_{ABS}^{REF}$ | 18 | 17 | 19 | 54 | 45 | 35 | 38 | 39 | -35 | -32 | -34 | -34 | 0.22 | 0.60 | 0.67 | 0.49 |
| MER$^{API2}$ *vs* TChl$a_{HPLC}^{REF}$ | 18 | 17 | 19 | 54 | 48 | 39 | 42 | 43 | -21 | -24 | -26 | -24 | 0.18 | 0.54 | 0.66 | 0.38 |
| MLP ($R_{rs}$$^{MER}$) *vs* TChl$a_{ABS}^{REF}$ | 18 | 17 | 19 | 54 | 23 | 32 | 30 | 29 | 8 | 8 | 16 | 11 | 0.69 | 0.51 | 0.85 | 0.67 |
| MLP ($R_{rs}$$^{MER}$) *vs* TChl$a_{HPLC}^{REF}$ | 18 | 17 | 19 | 54 | 66 | 45 | 49 | 54 | 39 | 16 | 30 | 29 | 0.38 | 0.49 | 0.49 | 0.43 |
| MLP ($R_{rs}$$^{SITU}$) *vs* TChl$a_{ABS}^{REF}$ | 93 | 91 | 113 | 297 | 16 | 17 | 19 | 17 | 3 | -4 | 7 | 2 | 0.88 | 0.91 | 0.91 | 0.91 |
| MLP ($R_{rs}$$^{SITU}$) *vs* TChl$a_{HPLC}^{REF}$ | 93 | 91 | 113 | 297 | 56 | 35 | 39 | 43 | 27 | 7 | 20 | 18 | 0.48 | 0.86 | 0.61 | 0.63 |

**Table 2. Comparison between the regional MLP($R_{rs}^{MER}$) and the standard MER$^{API2}$ (The location of ROIs is presented in Fig. 3e).**

| ROI | $N_{tot}$ | $N_{val}$ | $\varepsilon^*(\%)$ | $\delta^*(\%)$ | $r^2$ |
|---|---|---|---|---|---|
| #1 | 2122 | 2075 | 43 | -43 | 0.70 |
| #2 | 3383 | 1739 | 32 | -30 | 0.71 |
| #3 | 2946 | 2224 | 20 | -15 | 0.76 |
| **Total** | 8451 | 6038 | 32 | -29 | 0.76 |

**Table 3. List of notations.**

| | |
|---|---|
| API1 | Algal Pigment Index 1 |
| API2 | Algal Pigment Index 2 |
| BEAM | Basic ERS & ENVISAT (A) ATSR and MERIS Toolbox |
| BOA | Bottom-of-atmosphere |
| CDOM | Coloured Dissolved Organic Matter |
| Chl$a$ | Chlorophyll $a$ |
| EO | Earth Observation |
| $E_s(\lambda)$ | Surface downwelling incident irradiance |
| HPLC | High Performance Liquid Chromatography |
| $L_u(\lambda)$ | Subsurface upwelling radiance |
| $L_w(\lambda)$ | Water leaving radiance |
| MER$^{API2}$ | MERIS Algal Pigment Index 2 standard product |
| MERIS | MEdium Resolution Image Spectrometer |
| MLP | Multilayer Perceptron |
| MLP($R_{rs}{}^{MER}$) | Regional TChl$a$ products computed using inversion schemes based on the MLP NN using standard MERIS BOA $R_{rs}$ |
| MLP($R_{rs}{}^{SITU}$) | Regional TChl$a$ products computed using inversion schemes based on the MLP NN using in situ $R_{rs}$ |
| NN | Neural Nets |
| $N_{tot}$ | Number of total (i.e., $\eta < 1$) data points |
| $N_{val}$ | Number of valid (i.e., $\eta < 1$) data points |
| OLCI | Ocean Land Colour Instrument |
| PCA | Principal Component Analysis |
| $r^2$ | Coefficient of determination |
| ROIs | Regions of interest |
| $R_{rs}$ | Remote sensing reflectances |
| $R_{rs}{}^{MER}$ | Standard MERIS BOA $R_{rs}$ |
| $R_{rs}{}^{SITU}$ | In situ $R_{rs}$ |
| TChl$a$ | Total Chlorophyll $a$ |
| TChl$a_{ABS}^{REF}$ | API2 in situ equivalent algal pigment index |
| TChl$a_{HPLC}^{REF}$ | TChl$a$ concentration (monovinyl Chl$a$ + divinyl Chl$a$ + chlorophyllide $a$ + phaeopigments) determined by HPLC |
| $\delta$ | Signed percent differences |
| $\delta^*$ | Signed unbiased differences |
| $\varepsilon$ | Absolute percent differences |
| $\varepsilon^*$ | Absolute unbiased differences |
| $\eta$ | Novelty index |
| $\rho_N$ | Normalized water leaving reflectance |

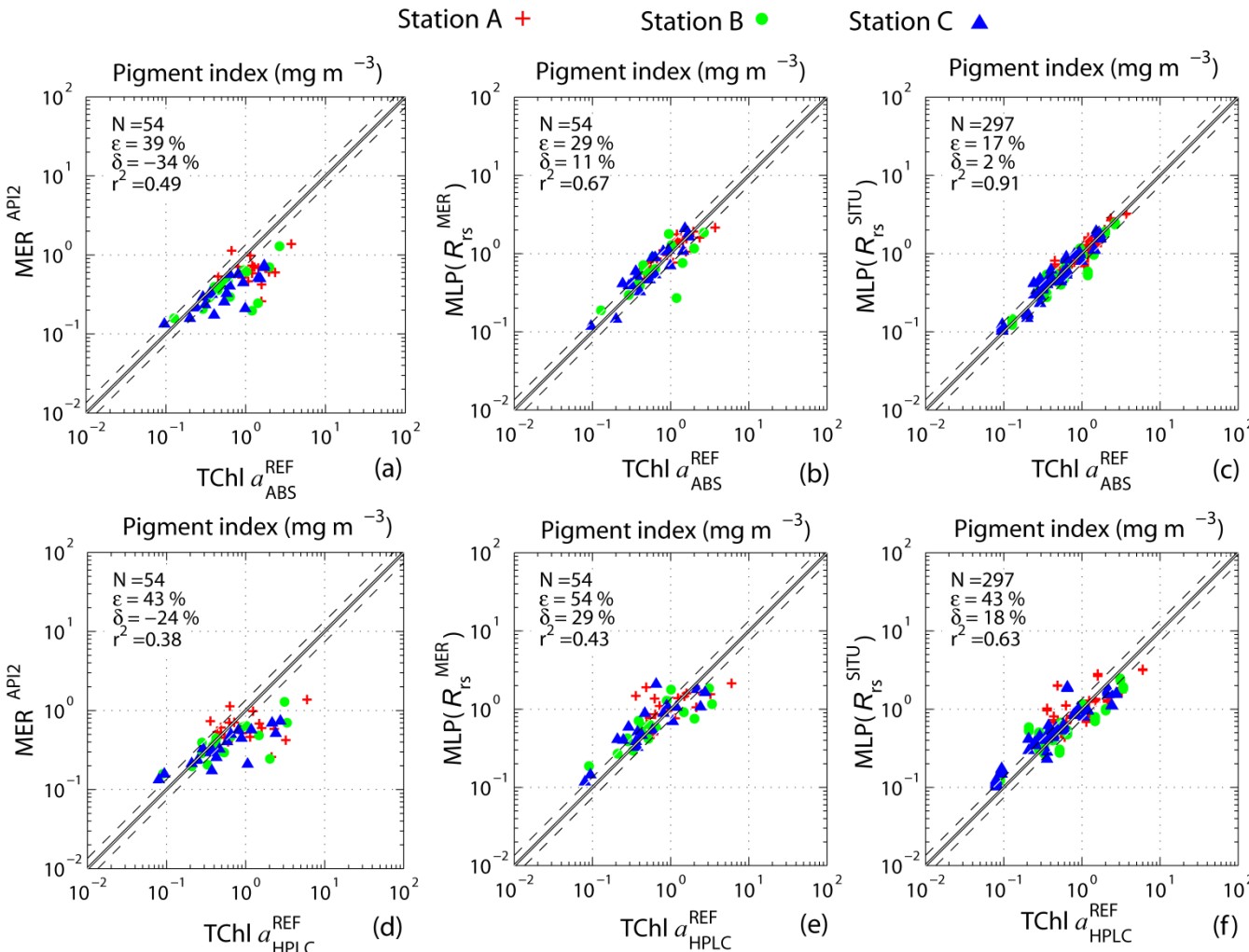

**Figure 1: Comparison between MERIS standard Algal Pigment Index 2 and results obtained by applying the Multilayer Perceptron (MLP) regional scheme for the Sagres region. The top row panels present the matchup comparisons with respect to the in situ reference TChl$a_{ABS}^{REF}$, while the lower panels detail the matchup comparisons with TChl$a_{HPLC}^{REF}$.**

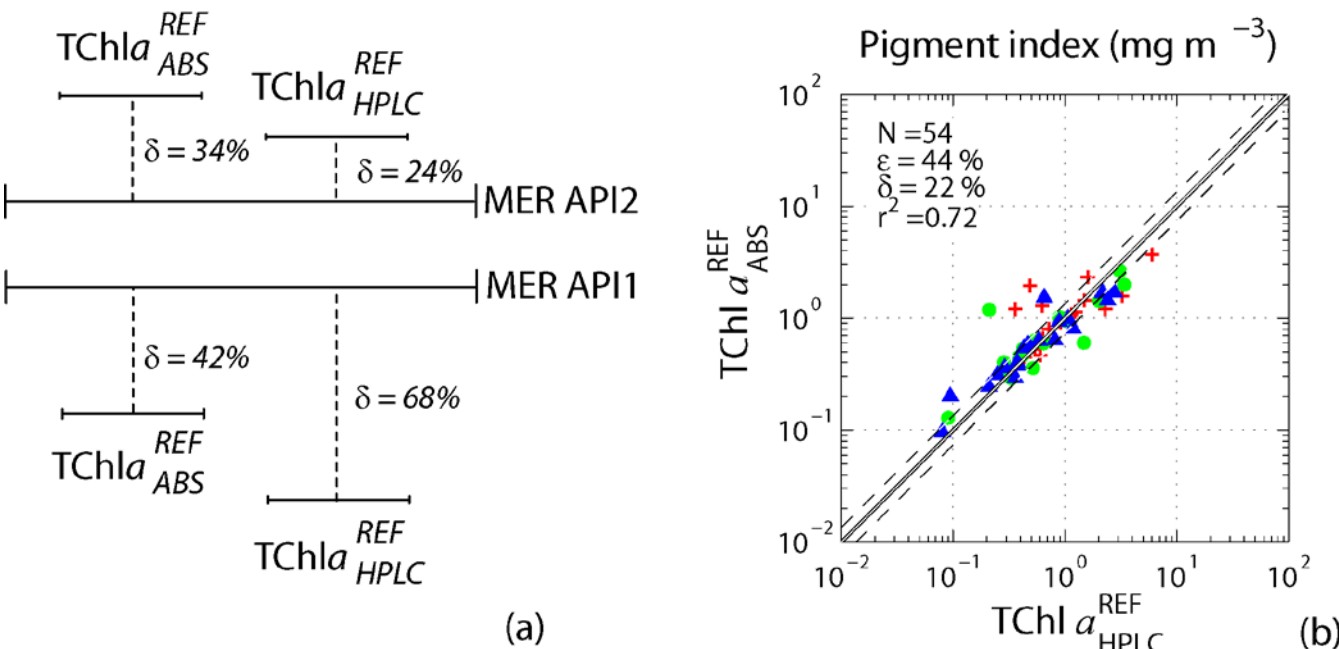

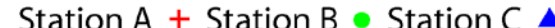

Figure 2: a) Schematic diagram showing, respectively, underestimation and overestimation of MERIS algal pigment indices 1 and 2, relative to TChl$a$, estimated through the absorption coefficient at 442 nm ( TChl$a_{ABS}^{REF}$) and measured by HPLC ( TChl$a_{HPLC}^{REF}$), and b) scatter plot of the TChl$a_{ABS}^{REF}$ *versus* TChl$a_{HPLC}^{REF}$ .

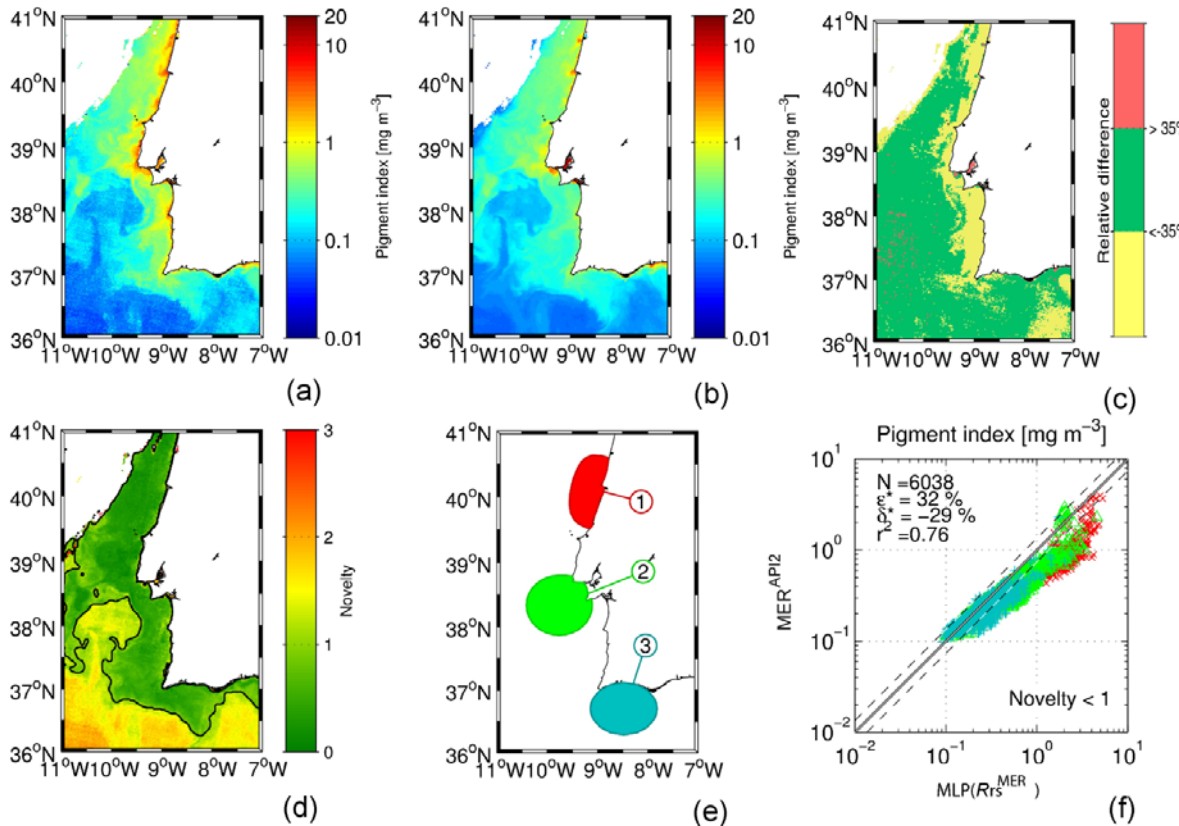

**Figure 3: Comparison between Sagres regional Multilayer Perceptron (MLP) algorithm map with MERIS pigment index product map Algal Pigment Index 2 for the 25th August 2010, showing a) the product map of the regional MLP, b) standard API2 MERIS product map, c) difference between MER$^{API2}$ and MLP($R_{rs}^{MER}$), d) region of applicability of MLP($R_{rs}^{MER}$), f) results of the application of the regional MLP to the Portuguese coast in the three regions of interest (shown in e). Please see Sect. 3.2 for a more detailed description of the panels. (Source: MER_RR_2PRAC20100825_103551_000026292092_00223_44365_0000.N1)**