# Peer review of "Technical Note: Algal Pigment Index 2 in the Atlantic off the Southwest Iberian Peninsula: standard and regional algorithms"

_Ocean Science, 2016_

## Referee Comment (RC1) · V. Suslin (Referee) · 9 Aug 2016

Questions

- page 3 line 30: you write " .. when $\eta$ is below the threshold $\eta$=1". Why $\eta$<1 but more or less 1?

- What the satellite data selection criteria you used in this study to analyze the quality of the Algal Pigment Index 2 in Sagres? It is clear that all was done 99 series * 3 (A, B, C) = 297 ground truth measurements, and the main factor is cloudy. What have any other criteria (except cloud) been used?

- Page 12: Fig. 1(c,f): Are you sure that in these figures N = 297? I think N = 54. Check

please!

- Page 13: In Fig. 2(b) I believe that you had the opportunity to show all measurements N = 297, not only N = 54!

General comments

This research is actual. The regional bio-optical algorithms demand for more reliable results by using satellite ocean color data. This study is a continuation of the work of these authors (eg, IEEE GEOSCIENCE AND REMOTE SENSING LETTERS, Digital Object Identifier 10.1109 / LGRS.2016.2529182) for the development of a regional satellite algorithm in the Atlantic off the southwestern Iberian Peninsula.

Specific comments

- in "Introduction" issue: To outline a significance of your research to add reference/references to other regional algorithms (for example, for the Mediterranean Sea) GREGG , W.W. and CASEY , H.W., 2004, Global and regional evaluation of the Sea-WiFS chlorophyll data set. Remote Sensing of Environment, 15 December, 93, Iss. 4, pp. 463–479, doi:10.1016/j.rse.2003.12.012.

- Statistics of in situ measurements by seasons is missed (among N=54/297 for stations A, B and C separatly). In particular, it could be useful in connection with Fig. 2b.

- Warning: "Tchla (Ref, ABS)" can not be equated with the concentration of chlorophyll a in Sagres, we can only speak of in situ aph (442). Do you agree?

Technical corrections

Page 3 line 2: "Total chlorophyll a (Tchla) .." repeat reference. The first reference to "Tchla" was on page 1 line 16 in Abstract.

Page 3 line 8: instead " .. neural nets NN" stay " .. neural nets" or " .. NN". The first reference to "NN" was on page 2 line 14

Page 3 line 8: "BOA" The first reference to "BOA" was on page 2 line 6

Page 3 line 16: http://ocportugal.org/sites/default/files/api2Sgr.pdf - Page not found

Page 3 line 7: remove ":" in "2.2.1 MERIS Standard algorithm API2:". The same for "2.2.1 Regional MLP NN algorithm:" on page 3 line 13

Page 4 line 1: "2.1 In situ reference data" move before issue "2.2.1 MERIS Standard algorithm API2".

Page 12: In Figure 1(a), a caption of the X axis can be seen partially

---

## Referee Comment (RC2) · Anonymous Referee #2 · 29 Aug 2016

As a technical note, this manuscript provides details the match up analysis between satellite retrieved estimates of chl-a and in situ measurements from different sources for a small region off the Iberian Peninsula. The results would be interesting to a limited readership who are interested in the same region. However I do think the paper has a major flaw; the authors find that the comparison of in situ chl-a parameters from different sources (absorption vs HPLC) yields better results than the comparison of retrieved parameters with either absorption or HPLC in situ results. These results are affected by the comparison of only 54 pairs of data for the retrieved vs in situ compared to 297 for the in situ abs vs HPLC. The potentially better metrics for the comparison of in situ parameters could be totally or in part due to the sample size being approximately

6x that of the retrieved vs in situ comparisons. I presume that the smaller data set is due to cloud cover etc so that you could only retrieve 54 data points that matched to an in situ measurement. If this is the case then the comparison between in situ abs vs HPLC should also only be for these same 54 data sets, so that all comparisons are being made on the same data sets.

Overall I think the idea of the paper is suitable as a technical note in OSD, but I would like to see the data and conclusions drawn after the authors re-analysed the data using the same 54 data sets for all comparisons, before I commented on the worth of the final paper.

Specific comments

A general comment is that there was a lot of acronyms and I think it would be useful to have a table which defined all the acronyms.

Pg 2, line 21: "of the Southwestern Iberian. . ." should be "off the Southwestern Iberian. . .."

Pg 3, line 8: delete neural nets and bottom-of-the-atmosphere as they have both already been defined

Pg 3, line 16: could not access web address provided – "page not found" message Pg 3, line 21: applicability should be application

Pg 3, line 23: applicability should be application

Pg 3, line 25: PCA should be in brackets – (PCA)

Pg 3, line 29: remove "novelty index" or "ÅŃ" as this term has already been defined

Pg 3, line 30: replace "when $\eta$ is below the threshold $\eta$=1." With "when $\eta < 1$."

Pg 4, line 3: replace "an hyperspectral" with "a hyperspectral"

Pg 4, line 3: delete "located below the surface" as it is implied by the preceding "subsurface".

Pg 4, line 10: replace "in GF/F" with "on GF/F"

Pg 4, line 14: the sodium hypochlorite bleaching does not remove the detrital contribution; it removes the pigment contribution. The phytoplankton contribution is determined as the difference between the total particulate and detrital absorption which are recorded before and after the hypochlorite bleaching, respectively.

Pg 7, line 4: "An additional explanation could be that TChlABSREF was determined using aph(442), which is likely better related to Rrs than TChlHPLCREF (both aph (442) and Rrs directly represent optical properties)." aph(442), might be better related to Rrs, but TChlHPLC is a direct measurement of the chl-a concentration whereas the aph(442) is an indirect measurement of the absorption due to phytoplankton. It is estimated as the difference between the total particulate and detrital absorption, both of which are measured, but would carry errors associated with the technique (extraction efficiency of the pigments, the dominance of a detrital signal etc) which would affect the accuracy of the estimation of aph(442).

Figures and captions

A general comment is that if the reader prints this publication, the font size used on the figures is quite small and can make reading difficult, especially both parts of Figure 2.

Figure 3: should have a description of each panel in the legend rather than referring to a section in the text. It is difficult to read both the section and the plot at the same time on a computer.

---

## Author Comment (AC1) · 30 Sep 2016

"Technical Note: Algal Pigment Index 2 in the Atlantic off the Southwest Iberian Peninsula: standard and regional algorithms" Manuscript Ref.: os-2016-41

Replies to Comments of Referee V. Suslin

Note from the authors: Please see the pdf supplement for the correct formatting of characters.

Questions Comments from Referee: page 3 line 30: you write " .. when _ is below the threshold _=1". Why _<1 but more or less 1?

Authors' response: The authors agree that this sentence was not easy to read, and have changed it accordingly. It should be: "...when $\eta \leq 1$".

Authors' changes in manuscript: In page 4, line 17, where it was "when $\eta$ is below the threshold $\eta=1$.", it is now "...when $\eta \leq 1$.".

Comments from Referee: What the satellite data selection criteria you used in this study to analyze the quality of the Algal Pigment Index 2 in Sagres? It is clear that all was done 99 series * 3 (A, B, C) = 297 ground truth measurements, and the main factor is cloudy. What have any other criteria (except cloud) been used?

Authors' response: The selection of satellite images was restricted to images without clouds and contamination, as indicated by not having specific Product Confidence (PCD) flags. The most common flags were PCD1_13 and PCD 19, where: PCD1_13 flag is a composite confidence flag for all the reflectance wavebands, and indicates a failure in the atmospheric correction for at least one of these wavebands and PCD 19 is a flag for uncertain aerosol type and optical thickness, i.e., also linked to the atmospheric correction. High levels of sun glint affected some of the days, and the corresponding flag was taken into account to check if the data were contaminated by a bright pattern of specular reflectance from the sun. An ice haze flag was also checked for some of the MERIS images when there was high radiance in the blue region of the spectrum caused by ice in the atmosphere or by a very high optical thickness. More details are in Cristina et al. 2014 and Cristina et al. 2016.

Authors' changes in manuscript: The authors have now included a section in Page 3, Line 2 about the image selection criteria, quoting "The selection of satellite images was restricted to images without clouds and contamination, as indicated by not having specific Product Confidence (PCD), sun glint and ice flags. More details on the image selection criteria and full description of flags are reported in Cristina et al. 2016 (Cristina, S., Cordeiro, C., Lavender, S., Goela, P.G., Icely, J., Moore, G., Newton, A. Remote Sensing. 2016. Seasonal-Trend decomposition time series based on Loess

applied to MERIS products from the SW Iberian Peninsula: Sagres. Remote Sensing, 8(6), 449; doi:10.3390/rs8060449.)"

Comments from Referee: Page 12: Fig. 1(c,f): Are you sure that in these figures N = 297? I think N = 54. Check please!

Authors' response: The authors thank the referee for noting this issue. In fact, there was an important detail requiring explanation in the manuscript. Regarding Fig.1, two different analyses are shown: a validation exercise (in left and middle panel) of MERIS products data against in situ reference data, and the other analysis (right panel) is the assessment of the performance of the regional NN algorithm for the retrieval of TChla. The different numbers of data points arise from the differences between the two analyses, the greater number of data points is used to evaluate the algorithm on the basis of its best performance (e.g. Cristina et al., 2016, Sá et al., 2015; Kajiyama et al., 2013). The x and y axes of the figures in the left and middle panels (Figs. 1a-d) represent the values of API2 product as retrieved by both MERIS and by the regional algorithm using MERIS reflectances, respectively. In these cases, the total numbers of points compared were 54. In contrast, Figs. 1c and 1f represent the regional algorithm trained using in situ reflectances collected from the in situ deployment of a Satlantic$^{®}$ radiometer. In this case, 4 to 8 reflectance casts were collected with the radiometer for each location corresponding to one in situ TChla measurement. As the objective was the regional algorithm performance assessment, all those points were used for this comparison, showing the best case scenario for the use of the regional algorithm. However, we can still show that comparison results remain consistent with the reported statistical values (Figure A1 in attachment) even when using only one radiometric cast per location (i.e., N=54 as in right panels of Fig. A1) to compare MLP(RrsSITU) with in situ references (TChla (ABS, HPLC)).

Figure A1 – Comparison of the performance of the regional NN algorithm results using only N=54 points (right panels), or with N=297 points (left panels, as originally Figs. 1c and 1f in the manuscript), both against TChla references (retrieved through aph(442)

and with HPLC).

Authors' changes in manuscript: A more detailed explanation has now been included in the manuscript to explain better the difference between number of data points, in Page 5, line 2, quoting: "The total number of samples used to validate MERAPI2 and MLP(RrsMER) algorithms results with respect to the in situ reference measurements was N=54. In contrast, the total number of samples for assessing the performance of regional MLP algorithm with in situ reference measurements (MLP(RrsSITU), was N=297. This larger number of samples is based on the data from 4-8 radiometric casts for each in situ TChla sample at each location."

Comments from Referees: Page 13: In Fig. 2(b) I believe that you had the opportunity to show all measurements N = 297, not only N = 54!

Authors' response: The authors thank the referee for noting this issue. In fact, there were important details requiring explanation in the manuscript. In Fig. 2b the two techniques for retrieval of reference TChla are compared. As explained by the authors in the previous comment, the number of in situ measurements for TChla retrieval (either through absorption or by HPLC) at surface was only 54. The number of samples was instead set to 297 in Figs 1c and 1f, because at each location sampled for TChla retrieval, 4 to 8 radiometric casts were collected. However the radiometric dataset is not represented in Fig. 2b, only in situ TChla measurements.

Authors' changes in manuscript: As mentioned in the previous comment, now the manuscript will include a more detailed explanation on the difference between the number of data points (Page 5, Line 2).

"GENERAL COMMENTS This research is actual. The regional bio-optical algorithms demand for more reliable results by using satellite ocean color data. This study is a continuation of the work of these authors (eg, IEEE GEOSCIENCE AND REMOTE SENSING LETTERS, Digital Object Identifier 10.1109 / LGRS.2016.2529182) for the development of a regional satellite algorithm in the Atlantic off the southwestern Iberian

Peninsula.

Specific comments"

Comments from Referees: in "Introduction" issue: To outline a significance of your research to add reference/ references to other regional algorithms (for example, for the Mediterranean Sea) GREGG , W.W. and CASEY , H.W., 2004, Global and regional evaluation of the SeaWiFS chlorophyll data set. Remote Sensing of Environment, 15 December, 93, Iss. 4, pp. 463–479, doi:10.1016/j.rse.2003.12.012.

Authors' response: The authors acknowledge the suggestion, and have now included this reference in the introduction section.

Authors' changes in manuscript: In page 2, Lines 12/13, the following sentence and references were added, quoting: "In such cases, regionalized bio-optical algorithms are required (Bricaud et al., 2002, Gregg and Casey, 2004)."

Comments from Referees: Statistics of in situ measurements by seasons is missed (among N=54/297 for stations A, B and C separatly). In particular, it could be useful in connection with Fig. 2b.

Authors' response: The authors agree with the referee on the utility of the analysis of the seasonality. Notwithstanding, the scope of this brief technical note was to evaluate the performance of a regional algorithm for the retrieval of TChla and also on the product definition itself. The analysis of seasonal components and trends would also imply the consideration of forcing agents (e.g. upwelling), which could be considered as an interesting follow up work, but a bit far from the scope of the present technical note.

Comments from Referees: Warning: "Tchla (Ref, ABS)" cannot be equated with the concentration of chlorophylla in Sagres, we can only speak of in situ aph (442). Do you agree?

Authors' response: In this technical note, the authors are discussing pigment indices derived from different quantities, having taken into account the definition of algal pig-

ment indices (API1 and API2) by the European Space Agency. API1 is equivalent to the concentration of TChla as determined by HPLC, and API2 is a proxy of TChla concentration determined by means of an empirical relationship between aph(442) and TChla. In page 3, lines 27/28, it is explained that TChla (Ref, ABS) is the in situ API2 equivalent measure estimated through aph(442), using the following expression: MERAPI2 = A $\times$ aph(442)B, where A=21.0 and B=1.04 (derived from field measurements in the German Bight and Norwegian waters as in Doerffer and Schiller, 2007). To ensure that the comparisons were the most reliable possible, the choice of the in situ references was made based on these definitions.

Comments from Referees (Technical corrections): Page 3 line 2: "Total chlorophyll a (Tchla) .." repeat reference. The first reference to "Tchla" was on page 1 line 16 in Abstract.

Authors' response: Thanks for noticing. The manuscript has been revised acknowledging the Referee's recommendation.

Authors' changes in manuscript: The Referee's request has been addressed in the revised manuscript, where it was "... Total chlorophyll a (TChla) . . ." (page 3 line 2), it has been changed to ". . .TChla . . ." now in page 3 line 4.

Comments from Referees (Technical corrections): Page 3 line 8: instead " .. neural nets NN" stay " .. neural nets" or " .. NN". The first reference to "NN" was on page 2 line 14.

Authors' response: Thanks for noticing. The manuscript has been revised acknowledging the Referee's recommendation.

Authors' changes in manuscript: The Referee's request has been addressed in page 3 in line 26 in the revised manuscript, where it was "... is estimated with two neural nets NN", and has now been changed to ". . .is estimated with two NN".

Comments from Referees (Technical corrections): Page 3 line 8: "BOA"The first reference to "BOA" was on page 2 line 6.

Authors' response: Thanks for noticing. The manuscript has been revised acknowledging the Referee's recommendation.

Authors' changes in manuscript: The Referee's request has been addressed in page 3 in line 26 in the revised manuscript, where it was "...bottom of the atmosphere (BOA)...", and has been changed to "...computes BOA...".

Comments from Referees (Technical corrections): Page 3 line 16: http://ocportugal.org/sites/default/files/api2Sgr.pdf - Page not found

Authors' response: Thanks for noticing. The revised manuscript has the corrected link.

Authors' changes in manuscript: The correction of the link in the revised manuscript was made in page 4 line 4, instead ..."http://ocportugal.org/sites/default/files/api2Sgr.pdf"..., and now is going to be "...http://ocportugal.org/sites/default/files/mlpSgrAPI2.pdf".

Comments from Referees (Technical corrections): Page 3 line 7: remove ":" in "2.2.1 MERIS Standard algorithm API2:". The same for "2.2.1 Regional MLP NN algorithm:" on page 3 line 13

Authors' response: Thanks for noticing. The manuscript has been revised acknowledging the Reviewer's recommendation.

Authors' changes in manuscript: The ":" were removed from "2.2.1 MERIS Standard algorithm API2:" and "2.2.1 Regional MLP NN algorithm:". These two sub-sections were changed in the revised manuscript to "2.2.1 MERIS Standard algorithm API2" in page 3 line 25 and "2.2.2 Regional MLP NN algorithm" in page 4 line 1.

Comments from Referees (Technical corrections): Page 4 line 1: "2.1 In situ reference data" move before issue "2.2.1 MERIS Standard algorithm API2".

Authors' response: This section has been moved following the referee's suggestion.

Authors' changes in manuscript: The Section "2.1 In situ reference data" in page 4 line 1 has now been moved to page 3, line 8.

Comments from Referees (Technical corrections): "Page 12: In Figure 1(a), a caption of the X axis can be seen partially." Authors' response: Thanks for noticing. The Figure 1(a) has been revised and the axis legend can now be seen in full (Fig 1 in attachment).

Authors' changes in manuscript: Fig. 1 was altered, to meet this requirement. The new figure is included.

Please also note the supplement to this comment:
http://www.ocean-sci-discuss.net/os-2016-41/os-2016-41-AC1-supplement.pdf
* * *
[Figure]

Station A +    Station B ●    Station C ▲

[Figure]

**Fig. 1.** Figure A1 Comparison between N=297 and N=54 (Full legend within the reply)

[Figure]

Fig. 2. Figure 1 - Correction to Fig.1 in the manuscript

[Figure]

---

## Author Comment (AC2) · 30 Sep 2016

"Technical Note: Algal Pigment Index 2 in the Atlantic off the Southwest Iberian Peninsula: standard and regional algorithms"

Manuscript Ref.: os-2016-41

Replies to Comments of Reviewer #2 (Anonymous Referee)

Note from the authors: Please see the Supplement pdf file of the reply for the correct formatting of the characters.

GENERAL COMMENTS

[Figure]

As a technical note, this manuscript provides details the match up analysis between satellite retrieved estimates of chl-a and in situ measurements from different sources for a small region off the Iberian Peninsula. The results would be interesting to a limited readership who are interested in the same region. However I do think the paper has a major flaw; the authors find that the comparison of in situ chl-a parameters from different sources (absorption vs HPLC) yields better results than the comparison of retrieved parameters with either absorption or HPLC in situ results. These results are affected by the comparison of only 54 pairs of data for the retrieved vs in situ compared to 297 for the in situ abs vs HPLC. The potentially better metrics for the comparison of in situ parameters could be totally or in part due to the sample size being approximately 6x that of the retrieved vs in situ comparisons. I presume that the smaller data set is due to cloud cover etc so that you could only retrieve 54 data points that matched to an in situ measurement. If this is the case then the comparison between in situ abs vs HPLC should also only be for these same 54 data sets, so that all comparisons are being made on the same data sets. Overall I think the idea of the paper is suitable as a technical note in OSD, but I would like to see the data and conclusions drawn after the authors re-analysed the data using the same 54 data sets for all comparisons, before I commented on the worth of the final paper.

Authors' response: The authors thank the referee for noting this issue. In fact, there was an important detail requiring explanation in the manuscript. Regarding Fig.1, two different analyses are shown: a validation exercise (in left and middle panel) of MERIS products data against in situ reference data, and the other analysis (right panel) is the assessment of the performance of the regional NN algorithm for the retrieval of TChla. The different numbers of data points arise from the differences between the two analyses, the greater number of data points is used to evaluate the algorithm on the basis of its best performance (e.g. Cristina et al., 2016, Sá et al., 2015; Kajiyama et al., 2013). The x and y axes of the figures in the left and middle panels (Figs. 1a-d) represent the values of API2 product as retrieved by both MERIS and by the regional algorithm using MERIS reflectances, respectively. In these cases, the total number of

points compared were 54. In contrast, Figs. 1c and 1f represent the regional algorithm trained using in situ reflectances collected from the in situ deployment of a Satlantic® radiometer. In this case, 4 to 8 reflectance casts were collected with the radiometer for each location corresponding to one in situ TChla measurement. As the objective was the regional algorithm performance assessment, all those points were used for this comparison, showing the best case scenario for the use of the regional algorithm. However, we can still show that comparison results remain consistent with the reported statistical values (Figure A1 in attachment) even when using only one radiometric cast per location (i.e., N=54 as in right panels of Fig. A1) to compare MLP(RrsSITU) with in situ references (TChla (ABS, HPLC)).

Figure A1 – Comparison of the performance of the regional NN algorithm results using only N=54 points (right panels), or with N=297 points (left panels, as originally Figs. 1c and 1f in the manuscript), both against TChla references (retrieved through aph(442) and with HPLC).

Authors' changes in manuscript: A more detailed explanation has now been included in the manuscript to explain better the difference between number of data points, in Page 5, line 2, quoting: "The total number of samples used to validate MERAPI2 and MLP(RrsMER) algorithms results with respect to the in situ reference measurements was N=54. In contrast, the total number of samples for assessing the performance of regional MLP algorithm with in situ reference measurements (MLP(RrsSITU), was N=297. This larger number of samples is based on the data from 4-8 radiometric casts for each in situ TChla sample at each location."

Specific comments

Comments from Referees: A general comment is that there was a lot of acronyms and I think it would be useful to have a table which defined all the acronyms.

Authors' response: The Referee's suggestion to create a table with all the acronyms was appreciated and included in the revised manuscript (Table 3).

Authors' changes in manuscript: Page 12 includes Table 3, with the list of the acronyms used in the manuscript.

Comments from Referees: Pg 2, line 21: "of the Southwestern Iberian..." should be "off the Southwestern Iberian..."

Authors' response: Thank you for noticing. The sentence has been modified in the revised manuscript.

Authors' changes in manuscript: In page 2, line 21, where it was "of the Southwestern Iberian...", has been changed to "...off the Southwestern Iberian...".

Comments from Referees: Pg 3, line 8: delete neural nets and bottom-of-the-atmosphere as they have both already been defined.

Authors' response: Thanks for noticing. The manuscript has been revised acknowledging the Reviewer's recommendation.

Authors' changes in manuscript: The Referee's request has been addressed in page 3 in line 26 in the revised manuscript, where it was "...bottom of the atmosphere (BOA)...", has been changed to "...computes BOA...".

Comments from Referees: Pg 3, line 16: could not access web address provided – "page not found" message

Authors' response: Thanks for noticing. The revised manuscript has the corrected link.

Authors' changes in manuscript: The correction of the link in the revised manuscript was made in page 4 line 4, where..."http://ocportugal.org/sites/default/files/api2Sgr.pdf"... has been changed to "...http://ocportugal.org/sites/default/files/mlpSgrAPI2.pdf".

Comments from Referees: Pg3, line 21: "applicability should be application"

Authors' response: Although other terminology could be applied, like the Referee's suggestion, the authors decided to maintain the same terminology, to assure the consistency with previously published studies (Cristina et al., 2016, Sá et al., 2015; Kajiyama et al., 2013) on similar topics.

Comments from Referees: Pg 3, line 23: "applicability should be application"

Authors' response: Although other terminology could be applied, like the Referee's suggestion, the authors decided to maintain the same terminology, to assure the consistency with previously published studies (Cristina et al., 2016, Sá et al., 2015; Kajiyama et al., 2013) on similar topics.

Comments from Referees: Pg 3, line 25: "PCA should be in brackets – (PCA)"

Authors' response: Thanks for noticing. The manuscript has been revised acknowledging the Referee recommendation.

Authors' changes in manuscript: Where it was "...Principal Component Analysis PCA ..." in the page 3, line 25, it has been changed in page 4 line 13 of the revised manuscript "...Principal Component Analysis (PCA)...".

Comments from Referees: Pg 3, line 29: remove "novelty index" or "ÅN′ " as this term has already been defined

Authors' response: Thanks for noticing this. The manuscript has been revised acknowledging the Referee's recommendation.

Authors' changes in manuscript: Where it was "...is its novelty index $\eta$ ..." in the page 3, line 29, has been changed in page 4 line 17 of the revised manuscript "...is its $\eta$ ...".

Comments from Referees: Pg 3, line 30: replace "when _ is below the threshold _=1." With "when _ < 1."

Authors' response: The authors agree that this sentence was not easy to read, and have changed the statement to: "...when $\eta \leq 1$".

Authors' changes in manuscript: In page 4, line 17, where it was "when $\eta$ is below the threshold $\eta$=1.", it is now "...when $\eta \leq 1$.".

Comments from Referees: Pg 4, line 3: replace "an hyperspectral" with "a hyperspectral"

Authors' response: Thanks for noticing this. The revised manuscript has replaced the word.

Authors' changes in manuscript: Where it was "...an hyperspectral..." in the page 4, line 3, has been changed to "...a hyperspectral ..."in page 3 line 10 of the revised manuscript.

Comments from Referees: Pg 4, line 3: delete "located below the surface" as it is implied by the preceding "subsurface".

Authors' response: Thanks for noticing this. The sentence was deleted following the Referee recommendation.

Authors' changes in manuscript: The sentence from the page 4 line 3 "...a subsurface radiance sensor Lu(ïĄň) located below the surface...", has been changed to "... a subsurface radiance sensor Lu(ïĄň)..." in page 3 line 8.

Comments from Referees: Pg 4, line 10: replace "in GF/F" with "on GF/F"

Authors' response: Thanks for noticing this. The word was changed following the Referee recommendation.

Authors' changes in manuscript: The word from the page 4 line 10 "...in GF/F...", has been changed to "...on GF/F..." in page 3 line 18.

Comments from Referees: Pg 4, line 14: the sodium hypochlorite bleaching does not remove the detrital contribution; it removes the pigment contribution. The phytoplankton contribution is determined as the difference between the total particulate and detrital absorption which are recorded before and after the hypochlorite bleaching,

respectively.

Authors' response: The authors agree that the sentence was not clear, and the manuscript will be changed accordingly.

Authors' changes in manuscript: In page 4, line 13/14, where the text "The phytoplankton absorption was determined from the total particle absorption, through the measurements before and after sodium hypochlorite bleaching of the filters to remove the contribution of detrital absorption" has been changed to "The phytoplankton absorption was determined as the difference between the total particulate and detrital absorption which were measured before and after sodium hypochlorite bleaching (Ferrari and Tassan, 1999; Goela et al., 2013), respectively.

Comments from Referees: Pg 7, line 4: "An additional explanation could be that TChlABSREF was determined using aph(442), which is likely better related to Rrs than TChlHPLCREF (both aph(442) and Rrs directly represent optical properties)." aph(442), might be better related to Rrs, but TChlHPLC is a direct measurement of the chl-a concentration whereas the aph(442) is an indirect measurement of the absorption due to phytoplankton. It is estimated as the difference between the total particulate and detrital absorption, both of which are measured, but would carry errors associated with the technique (extraction efficiency of the pigments, the dominance of a detrital signal etc) which would affect the accuracy of the estimation of aph(442).

Authors' response: The authors agree that this statement should be included in the manuscript.

Authors' changes in manuscript: This statement was added to the argument, in Page 7, line 10, quoting: "Some caveats would however apply to this argument, because ãÅŰTChlaãÅÙ_HPLCˆREF is a direct measurement of the TChla concentration whereas ãÅŰTChlaãÅÙ_ABSˆREF is an indirect measurement which has errors associated with the laboratorial determination of aph(442)".

Comments from Referees (Figures and captions): A general comment is that if the reader prints this publication, the font size used on the figures is quite small and can make reading difficult, especially both parts of Figure 2.

Authors' response: Thank you for noticing. The font size in the figure was expanded.

Authors' changes in manuscript: The figures in attachment have a larger font size.

Comments from Referees (Figures and captions): Figure 3: should have a description of each panel in the legend rather than referring to a section in the text. It is difficult to read both the section and the plot at the same time on a computer.

Authors' response: Thanks for the comment. The authors agree, and now a more detailed legend is presented.

Authors' changes in manuscript: The legend of Figure 3 was changed to: "Comparison between Sagres regional Multilayer Perceptron (MLP) algorithm map with MERIS pigment index product map Algal Pigment Index 2 for the 25th August 2010, showing a) the product map of the regional MLP, b) standard API2 MERIS product map, c) difference between MERAPI2 and MLP(RrsMER), d) region of applicability of MLP(RrsMER), f) results of the application of the regional MLP to the Portuguese coast in the three regions of interest (shown in e). Please see Sect. 3.2 for a more detailed description of the panels."

Please also note the supplement to this comment:
http://www.ocean-sci-discuss.net/os-2016-41/os-2016-41-AC2-supplement.pdf
* * *
Station A +    Station B ●    Station C ▲

Pigment index (mg m$^{-3}$)

N =297
ε = 17 %
δ = 2 %
r$^2$ =0.91

MLP($R_{rs}^{SITU}$)

TChl $a_{ABS}^{REF}$

Pigment index (mg m$^{-3}$)

N =54
ε = 17 %
δ = 3 %
r$^2$ =0.92

MLP($R_{rs}^{SITU}$)

TChl $a_{ABS}^{REF}$

Pigment index (mg m$^{-3}$)

N =297
ε = 43 %
δ = 18 %
r$^2$ =0.63

MLP($R_{rs}^{SITU}$)

TChl $a_{HPLC}^{REF}$

Pigment index (mg m$^{-3}$)

N =54
ε = 46 %
δ = 20 %
r$^2$ =0.65

MLP($R_{rs}^{SITU}$)

TChl $a_{HPLC}^{REF}$

**Fig. 1.** Figure A1 Comparison between N=297 and N=54 (Full legend within the reply)

[Figure]

[Figure]

**Fig. 2.** Figure 1 - Correction to Fig.1 in the manuscript

[Figure]

Station A ✛  Station B ●  Station C ▲

**Fig. 3.** Figure 2 - Correction to Fig. 2 in the manuscript

---

## Author Response (AR2)

**List of Alterations (relatively to the accepted manuscript)**

Dear Sirs,

The final version of the manuscript has minor alterations relative to the previous revised version (accepted for publication),

5  namely:

Fig.2 – The colours of some of the data points in Fig. 2b have been corrected, however these changes did not imply any change in the statistical analysis.

20

The following pages show the alterations, in the form of marked manuscript (alterations highlighted in yellow).

MARKED MANUSCRIPT

[revised manuscript text omitted]
_{\text{rs}}$ data ($R_{\text{rs}}^{\text{MER}}$). Corresponding data products are denoted $\text{MLP}(R_{\text{rs}}^{\text{SITU}})$ and $\text{MLP}(R_{\text{rs}}^{\text{MER}})$, respectively. In both cases, $R_{\text{rs}}$ at 490, 510 and 560 nm were selected as input channels, in agreement with the reference study (Cristina et al., 2014).

A novelty detection scheme (D'Alimonte et al., 2014; Bishop, 1994) was used to verify the algorithm applicability range by evaluating the representativeness of the input data in the training dataset (D'Alimonte et al., 2003; Mélin et al., 2011; Sá et al., 2015). The adopted applicability range is based on a novelty index (η) presented in published works (D'Alimonte et al., 2013; Sá et al., 2015). A revision is however applied for the scope of this work. This updated version considers all dimensions of the Principal Component Analysis (PCA) of selected input data, rather than only the first three components considered in the past (see the supplemental material for details). This updated definition is more effective for cases where the variability of training and application data tends to occur at different wavelengths (details not presented here). Key features are: 1) η is bounded between 0 and ∞; 2) the more the $R_{\text{rs}}$ spectrum is similar to the in situ MLP training measurements, the lower is its η; and 3) an $R_{\text{rs}}$ spectrum is considered within the MLP applicability range when η≤1.

**3 Results**

The main tasks of this study are: 1) to evaluate the performance of regional MLP algorithm and the $\text{MER}^{\text{API2}}$ results with respect to the in situ $\text{TChl}a_{\text{ABS}}^{\text{REF}}$ reference measurements; 2) to verify the applicability of the regional $\text{MLP}(R_{\text{rs}}^{\text{MER}})$ and to compare product maps with MER algal pigment indices; and 3) to extend the analysis by also considering $\text{TChl}a_{\text{HPLC}}^{\text{REF}}$ for data product assessment.

The statistical figures used to evaluate the estimated ($y$) in relation to the reference in situ TChl$a$ ($x$), were absolute (ε) and signed (δ) percent differences, defined as:

$$\varepsilon = \frac{1}{N}\sum_{i=1}^{N}\frac{|y_i - x_i|}{x_i} \times 100; \quad \delta = \frac{1}{N}\sum_{i=1}^{N}\frac{y_i - x_i}{x_i} \times 100 , \tag{2}$$

where $N$ is the total number of samples and $i$ is the sample index. For product maps comparison, the absolute ($\varepsilon^{*}$) and signed ($\delta^{*}$) unbiased differences were instead determined as:

$$\varepsilon^* = \frac{1}{N}\sum_{i=1}^{N}\frac{|y_i - x_i|}{y_i + x_i} \times 200; \quad \delta^* = \frac{1}{N}\sum_{i=1}^{N}\frac{y_i - x_i}{y_i + x_i} \times 200 \, , \qquad\qquad (3)$$

where $x_i$ and $y_i$ are the MLP($R_{\mathrm{rs}}{}^{\mathrm{MER}}$) and MER$^{\mathrm{API2}}$ values, respectively, taking the mean of the two values as a reference. In addition, the coefficient of determination $r^2$ between the evaluated quantities is also reported. The total number of samples used to validate MER$^{\mathrm{API2}}$ and MLP($R_{\mathrm{rs}}{}^{\mathrm{MER}}$) algorithms results with respect to the in situ reference measurements was N=54.

5 In contrast, the total number of samples for assessing the performance of regional MLP algorithm with in situ reference measurements (MLP($R_{\mathrm{rs}}{}^{\mathrm{SITU}}$), was N=297. This larger number of samples is based on the data from 4-8 radiometric casts for each in situ TChla sample at each location.

**3.1 Matchup data analysis**

The top panels of Fig. 1 present the matchup comparisons of MER$^{\mathrm{API2}}$, MLP($R_{\mathrm{rs}}{}^{\mathrm{MER}}$) and MLP($R_{\mathrm{rs}}{}^{\mathrm{SITU}}$) with respect to the in

10 situ reference TChl$a_{\mathrm{ABS}}^{\mathrm{REF}}$ (Figs. 1a, 1b and 1c, respectively). While MER$^{\mathrm{API2}}$ underestimated TChl$a$ ($\delta$ = -34%) especially at higher concentrations, the regional products slightly overestimated TChla ($\delta$ =11% for MLP($R_{\mathrm{rs}}{}^{\mathrm{MER}}$) and 2% for MLP($R_{\mathrm{rs}}{}^{\mathrm{SITU}}$). The best agreement between data sets was obtained with MLP($R_{\mathrm{rs}}{}^{\mathrm{SITU}}$), while MER$^{\mathrm{API2}}$ showed larger uncertainties. Table 1 presents the matchup analysis where the underestimation of MER$^{\mathrm{API2}}$ in relation to TChl$a$ is relatively constant (35%, 32% and 34%, in stations A, B and C, respectively) in all stations, but the correlation coefficient improves

15 with distance offshore (0.22, 0.60, 0.67 in stations A, B and C, respectively).

In general, the matchup analysis with TChl$a_{\mathrm{HPLC}}^{\mathrm{REF}}$ revealed higher uncertainties for MER$^{\mathrm{API2}}$, MLP($R_{\mathrm{rs}}{}^{\mathrm{MER}}$) and MLP($R_{\mathrm{rs}}{}^{\mathrm{SITU}}$), as detailed in Fig. 1 (lower panel). Note that also in this case MLP($R_{\mathrm{rs}}{}^{\mathrm{SITU}}$) presented the best results, with the highest coefficient of determination and the lowest bias. Similar to what was documented for TChl$a_{\mathrm{ABS}}^{\mathrm{REF}}$, the bias for TChl$a_{\mathrm{HPLC}}^{\mathrm{REF}}$ displayed only small differences between the sampling stations. The coefficient of determination instead increased from

20 station A to station C. The underestimation of MER$^{\mathrm{API2}}$ in relation to TChl$a_{\mathrm{HPLC}}^{\mathrm{REF}}$ was also observed, but with a lower bias (Fig. 1d). These observations are schematized in Fig. 2, where MER$^{\mathrm{API2}}$ was considered as the baseline. A complementary comparison with MER$^{\mathrm{API1}}$ is also presented for completeness. Results indicated an overestimation by the API1 algorithm in relation to both estimations of TChl$a$ (details not shown). The tendency of TChl$a_{\mathrm{ABS}}^{\mathrm{REF}}$ to produce higher values than TChl$a_{\mathrm{HPLC}}^{\mathrm{REF}}$ was also confirmed.

25 ### 3.2 Comparison of product maps

[revised manuscript text omitted]